# Defect Study and Modelling of SnX3-Based Perovskite Solar Cells with SCAPS-1D

**DOI:** 10.3390/nano11051218

**Published:** 2021-05-05

**Authors:** Md. Samiul Islam, K. Sobayel, Ammar Al-Kahtani, M. A. Islam, Ghulam Muhammad, N. Amin, Md. Shahiduzzaman, Md. Akhtaruzzaman

**Affiliations:** 1Department of Electrical and Electronic Engineering, Southeast University, Dhaka 1207, Bangladesh; sami.seu@gmail.com; 2Solar Energy Research Institute, The National University of Malaysia, Bangi 43600, Malaysia; 3Institute of Sustainable Energy, Universiti Tenaga Nasional (@The National Energy University), Kajang 43000, Selangor, Malaysia; ammar@uniten.edu.my; 4Department of Electrical Engineering, Faculty of Engineering, University of Malaya, Kuala Lumpur 50603, Malaysia; aminul.islam@um.edu.my; 5Department of Computer Engineering, College of Computer and Information Sciences, King Saud University, Riyadh 11461, Saudi Arabia; ghulam@ksu.edu.sa; 6Nanomaterials Research Institute, Kanazawa University, Kakuma, Kanazawa 920-1192, Japan; shahiduzzaman@se.kanazawa-u.ac.jp

**Keywords:** perovskite, CH_3_NH_3_SnBr_3_, solar cell, amphoteric defect, donor density, SCAPS

## Abstract

Recent achievements, based on lead (Pb) halide perovskites, have prompted comprehensive research on low-cost photovoltaics, in order to avoid the major challenges that arise in this respect: Stability and toxicity. In this study, device modelling of lead (Pb)-free perovskite solar cells has been carried out considering methyl ammonium tin bromide (CH_3_NH_3_SnBr_3_) as perovskite absorber layer. The perovskite structure has been justified theoretically by Goldschmidt tolerance factor and the octahedral factor. Numerical modelling tools were used to investigate the effects of amphoteric defect and interface defect states on the photovoltaic parameters of CH_3_NH_3_SnBr_3_-based perovskite solar cell. The study identifies the density of defect tolerance in the absorber layer, and that both the interfaces are 10^15^ cm^−3^, and 10^14^ cm^−3^, respectively. Furthermore, the simulation evaluates the influences of metal work function, uniform donor density in the electron transport layer and the impact of series resistance on the photovoltaic parameters of proposed n-TiO_2_/i-CH_3_NH_3_SnBr_3_/p-NiO solar cell. Considering all the optimization parameters, CH_3_NH_3_SnBr_3_-based perovskite solar cell exhibits the highest efficiency of 21.66% with the *V_oc_* of 0.80 V, *J_sc_* of 31.88 mA/cm^2^ and Fill Factor of 84.89%. These results divulge the development of environmentally friendly methyl ammonium tin bromide perovskite solar cell.

## 1. Introduction

An undisputed revolution in the development of photovoltaic technology was achieved by solar cells based on metal halide perovskites with the formula of ABX_3_ (where A is an organic or inorganic cation, B is a lead or tin cation and X is a halide anion). Over the past few years, the power conversion efficiency (PCE) of perovskite solar cells (PVSCs) has increased rapidly from 3.9% to a certified 22.7% [1,2,3,4,5]. The remarkable increase in PCE is attributed to the supreme optoelectronic properties, such as high absorption coefficient (~10^5^ cm^−1^), low exciton binding energy (~20 meV), and relatively long diffusion duration of the carrier (>1 μm) [6,7,8,9]. In addition, perovskite structure engineering from the simple methylammonium lead iodide (CH_3_NH_3_PbI_3_) perovskite to the new mixed-cation and mixed-anion halide perovskite materials also results in high efficiency [5,10,11,12]. Despite the rapid development, to achieve stability and meet the Shockley-Queisser Limit (SQL), which is ~30.5% PCE for a single Methylammonium Lead Iodide (MAPbI_3_)-based junction cell, the overall PCE of perovskite-based solar cells is still far away. Again, it is already reported that NiO-based inverted structure device can demonstrate superior stability than planner structure [13]. Therefore, it is particularly important, not only to minimize environmental toxicity by removing Pb in the perovskite composition, but also to achieve stability by compositional engineering without compromising photovoltaic efficiency to facilitate the commercialization of PVSCs [14,15,16]. The partial substitution or full replacement of Pb with comparatively less toxic tin (Sn) is an obvious choice among the many approaches investigated because of the similarity in their ionic radii (Sn^2+^: 0.93 Å vs. Pb^2+^: 1.20 Å) and electronic configurations that could theoretically retain the exceptional semiconducting properties of APbX_3_ [14]. In comparison to their pristine Pb-based counterparts, the synthesis of Sn-based perovskites promotes the recognition of lower band gap (Eg) perovskites [17,18,19,20]. Again, researchers found that the tuning of halide ions in the perovskite structure can lead to higher stability of the device. In this respect, several other attempts have been made to build efficient low Eg PVSCs based on Sn [10,12,14]. Nevertheless, there are still many problems that hinder the production of Sn-based PVSC with bromine (Br) as cation. In order to develop a non-toxic, highly efficient and stable PVSC, this article proposes a new absorber layer (Figure 1) of perovskite-CH_3_NH_3_SnBr_3_.

Hence, a numerical simulation has been performed through SCAPS-1D software on a proposed new perovskite structure to validate its design and to evaluate its performance for commercial implementation. Moreover, amphoteric defect state of the absorber, defect tolerance at interfaces and electrical properties of the solar cell device have also been investigated.

## 2. Theoretical Concept

In nature, the perovskite structure (*ABX*_3_) is symmetric, where *A* is greater than *X* (Figure 1). *X* can be substituted with various O, Cl, Br, I, and S elements throughout the composition. Atom placements in the three-dimensional (3D) structure are primarily considered to be connected to the stability required for a chemical charge neutralization process [21]. A key requirement for perovskite minerals is that, while the basic structure appears plain, structural distortion characteristics are quite different [22]. Due to this, the solar cell researchers have been investigating Tin (Sn)-based perovskite solar cell as an alternative, where the lead is substituted by tin and modified material compositional structure looks like *A*Sn*X*_3_. The primary benefits of tin-based perovskite solar cells are that they are lead-free and can help tune the active layer’s bandgap further. Additionally, by reducing the acceptor doping the concentration of the active layer, the efficiency of the tin-based solar cell can be improved.

Another noteworthy point is that the incorporation of suitable cations to form perovskite structure are estimated based on two important parameters: Goldschmidt tolerance factor (*t*), and the octahedral factor (*μ*). Tolerance factor (*t*) is the function of iconic radii and a dimensionless parameter. Octahedral factor (*μ*) demonstrates the relation between ionic radii of *B* cation and *X* anion [23]. They are expressed as following equations,
(1)$$t=\frac{r_{A}+r_{X}}{\sqrt{2}\;\left(r_{B}+r_{X}\right)}$$
(2)$$µ=\frac{r_{B}}{r_{X}}$$
where *r_A_* and *r_B_* represents the ionic radii of two big size cations *A* and *B*, and *r_X_* represents the ionic radius of smaller halide anion. The tolerance factor (*t*) should fall within the range of 0.8–1.0 for the formation of robust perovskite structures. The variety of octahedral factor (*μ*) must be limited within 0.44–0.72 to form a stable *BX*_6_ octahedron for *B* cation and *X* anion. [24] Goldschmidt’s tolerance factor (*t*) has played a major role in the development of perovskites [25] and has also been used to develop/synthesize new hybrid organic-inorganic stable perovskite structures by formulating the composition of perovskites. The tolerance factor can be tailored to the stable perovskite range by combining different *A/B* cations and *X* anions in a particular composition [26,27,28,29].

In our proposed perovskite structure (CH_3_NH_3_SnBr_3_), *r_A_*, *r_B_* and *r_X_* values are 217, 110 and 181 pm respectively [30]. From Equation (1) and Equation (2), it is found that tolerance factor *(t)* and octahedral factor *(μ)* of proposed structure are 0.967, and 0.607 respectively, both are within the ideal range of stable perovskite structure. Hence, it can be said that the proposed perovskite structure (CH_3_NH_3_SnBr_3_) can be an ideal candidate for Pb free highly efficient stable perovskite solar cell.

## 3. Device Structure and Simulation Parameters

Besides the experimental study, the simulation environment provides a power full tool to better understand about the physical behavior of different optoelectronic properties of any solar cell. In relation to the electrical simulation, the software like the Solar Cell Capacitance Simulator (SCAPS-1D) [31,32,33,34,35,36,37,38] was used, which basically works on two basic semiconductor equations, including, the Poisson equation and the continuity equation of electrons and holes under steady-state condition.

In this study, a simulation based on hypothetical research has been performed to sketch the performance of lead (Pb) free Tin (Sn) based perovskite material which acts as an absorber layer, sandwiched by two transport layers- TiO_2_, and NiO respectively. The architecture of the device is FTO/TiO_2_/CH_3_NH_3_SnBr_3_/NiO, where FTO acts as the Transparent Conductive Oxide (TCO), TiO_2_ and NiO acts as the ETL, and HTL materials, respectively. The schematic structure and energy band diagram of this work are given below in Figure 2. 

For the incident radiation, a regular AM1.5 G illumination spectrum (1000 Watt/m^2^; T = 300 K) was employed. Standard layer thickness of were acquired from the various recorded articles [32,33,34]. In order to assess the consequence of defect densities subsisting on material boundaries (the perovskite/ETL and perovskite/HTL boundaries), a very thin interface defect layer of 10 nm was used. Additionally, in this simulation, the amphoteric inherent defect model is used to reproduce the defects of PSC, where the density of the defect in the active layer varied. Gaussian energy distribution with a characteristic energy of 0.1 eV was considered for all defect states. The thermal velocity of the electrons and holes of 1 × 10^7^ cm/s were taken during simulation. This simulation software was used to investigate the different performance metrics Fill factor (*FF*), short-circuit current density (*Jsc*), open-circuit voltage (*Voc*), efficiency (η) of a perovskite based solar cell [39]. Table 1 summarizes the material parameters that were used in this simulation.

## 4. Results and Discussion

### 4.1. Effect of Amphoteric Defect Density in the Absorber Layer

The CH_3_NH_3_SnBr_3_ layer exhibits various forms of conductivity based on the Sn and halide molecules deposition properties. Moreover, Sn^2+^ cataion is very prone to oxidation during the process of fabrication, which attributes defects inside the perovskite structure [35,40]. In order to determine the electron-hole diffusion length and open circuit voltage (*Voc*), defect properties in the solar cell absorber layer play a significant role. The definition of amphoteric defects was initially familiar with the clarifying effects on the properties of semiconducting materials of native (or inherent) defects. More recently, the principle of amphoteric native defects has been used in PSC devices to regulate the inclusion of defects in compound semiconductors, such as CH_3_NH_3_PbX_3_ [41]. In this work, amphoteric defect has been considered above the Ev of the absorber layer with uniform energetic distribution where defect state has been varied from 10^13^ cm^−3^ to 10^17^ cm^−3^. The details have been given at Appendix A (Table A1). Figure 3 exhibits the influence of amphoteric defect on solar cell parameters of proposed CH_3_NH_3_SnBr_3_ solar cell. It is found that *Voc* decreases gradually with the rise of defect density (Figure 3a). On the other hand, we did not observe any significant changes in short circuit current density (*Jsc*) and Fill Factor (*FF*) (Figure 3b,c) until density of defect state increased above 10^16^ cm^−3^. However, both parameters exhibited declining behavior beyond this critical value, which supports the theoretical aspect of the device performance. With the increase of defect densities, the unwanted recombination rate increased as the defects created the dangling bonds, which act like the trap state for the photo-generated charge carriers. This is the reason why the short circuit current decreased, which is eventually responsible for lowering the Fill Factor (*FF*) of the device. The device efficiency also dropped significantly from 24.5% to 16.17%, when amphoteric defect density increased to 10^15^ to 10^16^ cm^−3^. Such a sharp fall in solar cell performance can be attributed to the increasing number of recombination.

Figure 3d shows the typical relationship between *Voc, Jsc* and efficiency for different state of amphoteric defect densities. It was found that the decrease in efficiency is mainly due to the fall in *Jsc*. The defects act like the recombination centre of the photogenerated carriers, limiting the short circuit current (*Jsc*). Therefore, the defect tolerance for the tin (Sn)-based halide perovskite is around 1 × 10^15^ cm^−3^.

### 4.2. Effect of Interface Defect States 

The impact of defect states in both the interfaces of Perovskite Solar Cell (PSC), TiO_2_/MASnBr_3_ and MASnBr_3_/NiO, have been studied in detail. At the ETL/Perovskite interface, density of defect states has varied from 10^12^ cm^−3^ to 10^20^ cm^−3^ where the position of the defect state was considered at the above of *E_v_* with gaussian energetic distribution. It has been noted that *V_oc_* and efficiency decreases sharply (Figure 4a) from 0.99 V to 0.89 V, and 26.75% to 19.75%, respectively, when the density of the defect state in the interface reached 10^14^ cm^−3^. However, we did not find a notable observation for *Jsc* and *FF*, though both decreases (Figure 4b,c) with the increase of defect state density. Figure 4d expresses the correlation between *V_oc_, J_sc_* and defect states in the ETL/Perovskite interface. It can be observed that increments in the level of defect state mainly impact the open circuit voltage rather than short circuit current. Therefore, a change in *V_oc_* is mainly attributable to the drop in the efficiency of the device. Hence, defect tolerance at TiO_2_/MASnBr_3_ can be figured up to 10^14^ cm^−3^ as beyond this level efficiency of CH_3_NH_3_SnBr_3_ solar cell deteriorates to a great extent.

On the other hand, at the HTL/Perovskite interface, density of defect states has been varied from 10^13^cm^−3^ to 10^18^ cm^−3^ where position of defect state has been considered at the middle of the interface with uniform energetic distribution. To determine the efficiency of the overall structure, the defect densities of the whole transport layer and the perovskite absorber play a pivotal role. It has been revealed that *Voc* decreases steadily (Figure 5a) with the increase of defect density at HTL/Perovskite interface. In the case of *Jsc*, it decreased with the increase in the defect state (Figure 5b), and exhibited a sharp fall of current (24.18 mA/cm^2^ to 19.68 mA/cm^2^) when the density of the defect state increased from 10^15^ to 10^16^ cm^−3^. However, the change in fill factor with the increase of defect density (Figure 5c) was very negligible. However, the efficiency of the solar cell showed a steady decrease from 23.6% to 18.49% up to the defect density range 10^15^ cm^−3^. After that, there was a sharp decrease in efficiency of the cell with the increasing values of defect state. Figure 5d represents typical relationship between *Jsc* and *Voc* w.r.t defect density. The density of the defect state in the HTL/Perovskite interface was observed to primarily affect the short-circuit current than open circuit voltage when density of defect becomes over 10^14^ cm^−3^. The interface between the perovskite and NiO layer plays a vital role damaging the short circuit current. It. The critical issue regarding NiO thin film is the presence of surface defects acting as trap states in the solar cell structure, which seriously affects the charge carrier transfer since charge extraction only occurs at the interfaces. As a result, detrimental hysteresis and light soaking takes place due to charge recombination [42].

In NiO films, the defects are characteristically existing in the form of hydroxyl groups originating from the oxygen deficiency or unconverted Ni (OH)_2_ [43,44,45,46]. Hence, we can determine that tolerance of defect state for MASnBr_3_/NiO interface is 10^14^ cm^−3^.

### 4.3. Metal Work Function

The work function of a metal is the amount of energy or photons required to extract an electron from the metal surface [47]. It was reported that higher values of work function lead to increased solar cell efficiency [48,49]. This is attributed to the fact that the barrier height of the majority carrier decreases with the increase in hte work function value, which eventually makes contact more ohmic type. Therefore, as the metal’s work function increases, both the open circuit voltage and cell efficiency also increase. Au and Pt are the most commonly used back contact metal in solar cell but both are expensive. In this work, simulations were performed to identify a suitable earth abundant metal for using as back contact in the proposed device structure (Figure 2). The parameters for the contact materials used for this simulation are shown in Table 2 [47,49]. The efficiency of the solar cell CH_3_NH_3_SnBr_3_ (without defects) versus various metals (metal working function) used in the back contact of the system structure is shown in Figure 6 and Appendix B
Table A2.

### 4.4. Effect of Doping Density of the TiO_2_/MASnBr_3_ Heterojunction

Mott-Schottky (MS) is a well-known and effective instrument used assessing the built-in potential (V_bi_)- the difference between the functions of electrode operation [50] and the doping level of a device. MS theory is mainly based on the p-n junction properties [51], but it is also used in organic devices as well [52,53,54]. In the MS plot, slope of 1/C^2^ (V) interprets as a concentration of occupied trapping centers [53,54,55] and the *x*-axis intercept is usually representing the V_bi_ of organic semiconductor devices. Although the obtained values are lower than expected from the difference of the electrode work functions [56,57,58]. In reverse polarization, the capacitance of the system is the sum of the capacitance of the junction and the capacitance of the contact (for metal/TiO_2_ interface) [47,57]. The C-V characteristics and Mott-Schottky plot analysis of proposed solar cells were simulated in Figure 7a,b as a function of the TiO_2_ structure’s shallow uniform donor density (N_d_). The concentration of donor density (N_d)_ varied in the range from 10^14^ 1/cm^3^ to 10^18^ 1/cm^3^ keeping other variables as constant. It has been observed (Figure 7a) that the capacitance increases gradually with the applied voltage and increased sharply at higher voltage and reached a maximum when N_d_ is 10^18^ 1/cm^3^. From Figure 7a, it is clear that, at zero bias, this structure is fully depleted, but when the forward bias is applied, around 0.5 V, the depletion width shrinks to approximately equal to the thickness of the absorber layer. Therefore, capacitance increases with further increase in forward bias voltage, and behave according to Mott–Schottky relationship. It was already reported that the current is significantly lesser than the saturation current at low voltages, but the current was restricted to the saturation current at the contact at high voltages [59]. The lower value of the built-in potential (Vbi), obtained from the Mott-Schottky plot under illumination, can be ascribed to the capacitance originating from the photogenerated charge carriers that can build up in the low-mobility materials, even at reverse bias [60]. When doping concentration increases, charge accumulation increases at the interface and the capacitance value will also rise. Eventually the thickness of the depletion layer/space charge layer decreases, which eventually decrease the built-in potential (Vbi), according to the conventional capacitance equation:(3)$$C=\epsilon_{0}\frac{A}{d}.$$

This leads to decrease in the voltage at the interface of metal/TiO_2_. Figure 7b represents the determination of built-in potential (V_bi_) Mott– Schottky relation at 1/*C*^2^ = 0 on the potential axis [61]. The built-in potential (V_bi_) has been found to increase from 0.69 V to 0.77 V with the increase of Nd.

Flat band potential at the interface is an important parameter in designing solar cell. Larger flat band potential makes it easier for the charge carriers to transfer at the interface—which can be obtained from the Mott-Schottky plot using the following equation,
(4)1C2=[2qϵsϵ0Nd][Vapp−VFB−(KBTq)]
where *q* is charge on the carriers, *ɛ_s_* the permittivity of semiconductor, *ɛ*_0_ the permittivity of free space, N_d_ the donor density, V_FB_ the flat band potential, *K_B_* the Boltzmann’s constant and *T* the temperature of operation [62].

When the donor concentration increases the band bending increases, which lower the accumulation of charge carriers at the depletion zone and capacitance value decreases.

### 4.5. Optimized Device and Effects of Series Resistances on the Performance of the Solar Cell

Non-ideality in a solar cell is an established reality now, which hinders the performance of the cell. There are several factors responsible for this, and one of them being parasitic resistances. Both series (Rs) and shunt resistances reduce the output of a solar cell. Figure 8 represents the equivalent circuit of a solar cell. Prior to analyzing the effects of series resistance, we have performed the simulation based on defect tolerances, optimized metal work function (MWF) and ideal N_d_ that has been investigated above. It is found that proposed device exhibits 21.66% efficiency with *V_oc_* = 0.80 V, *J_sc_* = 31.88 mA/cm^2^ and *FF* = 84.89% (Figure 9a).

In the perovskite solar cells, basically a p-i-n or n-i-p structure is considered. Rs in solar cell mainly exists in the contacts or interfaces: resistance at HTL/perovskite interface, resistance at Electron Transport layer (ETL)/perovskite interface and resistance of the top and rear metal contacts. Moreover, when solar cells are exposed to climate, in practical applications inside a module, the thermomechanical fatigue or cracks develop in the solder bonds depending on the climatic conditions. These cracks lead to increase the Rs of solar cell.

It is already reported that increase in Rs affects directly to the drop of Fill Factor drop due to solder bond degradation, whereas the *J_sc_* drop is attributed to the optical transmission loss caused by the encapsulant discoloration [63]. The simulation results on varying series resistances (R_s_) on the proposed CH_3_NH_3_SnBr_3_ based solar cell structure is shown in Table 3 and figures are shown in Appendix C (Figure A1).

It has been observed that the increase in series resistance adversely affects the Fill Factor, and once Rs is remarkably high, it slightly affects the *J_sc_*. These findings are completely agreement with the reported literature. It has been observed that *FF* degrades almost 3.6% with each 0.01 Ω increase in Rs, which is a little higher than conventional Si (approximately 2.5% per 0.01 Ω increase in Rs) solar cell. However, it is encouraging that while *FF* degrades at 3.6%, but efficiency degrades much lower rate-only at 0.857% with 0.01 Ω increase in Rs.

## 5. Conclusions

In summary, this article unveiled a Pb free perovskite solar cell by non-toxic Sn based CH_3_NH_3_SnBr_3_ and analyzed with SCAPS 1D simulation software. The model was initially verified by Goldschmidt tolerance factor and the octahedral factor, and found to be in the ideal range of stable perovskite structure. As a native type defect, amphoteric defect is likely to form during fabrication process, thereby reducing its concentration to 10^15^ cm^−3^ can lead to a highly efficient solar cell. Again, controlling the defects at the interfaces are most critical factor for high efficiency of solar cells, and scaling down to as low as 10^14^ cm^−3^ in both the interfaces can significantly increase the PCE of the device. The study also suggests that the efficiency of n-TiO_2_/i-CH_3_NH_3_SnBr_3_/p-NiO solar cells can increase by replacing costly Au with Cu doped C as back contact material, and by selecting the optimal density state of the n-TiO_2_ layer. Again, the study portrays that while *FF* of proposed cell decreases approximately by 3.6% with the increase of 0.01Ω Rs, its effect on overall efficiency is truly minor. Based on optimization, the highest efficiency of 21.66% has been achieved for TiO_2_/i-CH_3_NH_3_SnBr_3_/p-NiO solar cell (*V_oc_* = 0.80 V, *J_sc_* = 31.88 mA/cm^2^ and *FF* = 84.89%), which is very promising compared to Pb-based perovskite solar cell. Nevertheless, this work successfully demonstrates the low-cost non-toxic CH_3_NH_3_SnBr_3_ based perovskite solar cell as a potential candidate in the photovoltaic industry.

## Figures and Tables

**Figure 1 nanomaterials-11-01218-f001:**
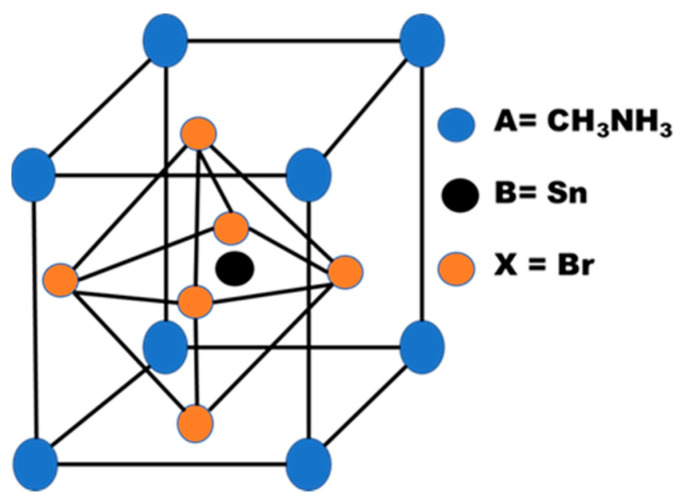
Proposed Perovskite Structure CH_3_NH_3_SnBr_3_.

**Figure 2 nanomaterials-11-01218-f002:**
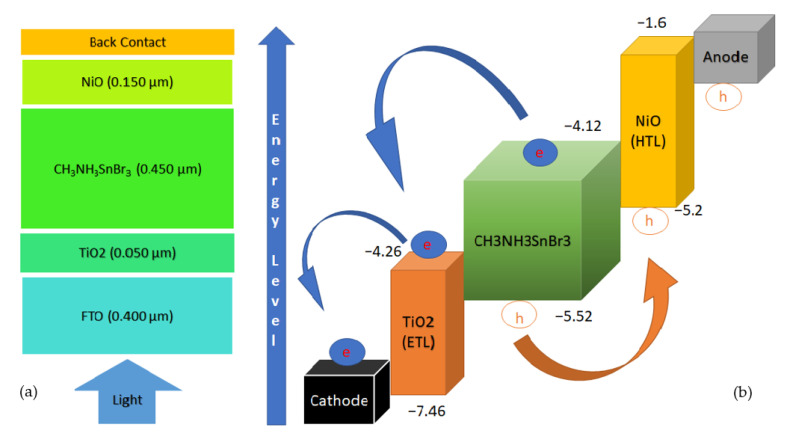
(**a**) Schematic diagram of proposed solar cell; and (**b**) energy band diagram of CH_3_NH_3_SnBr_3_ solar cell.

**Figure 3 nanomaterials-11-01218-f003:**
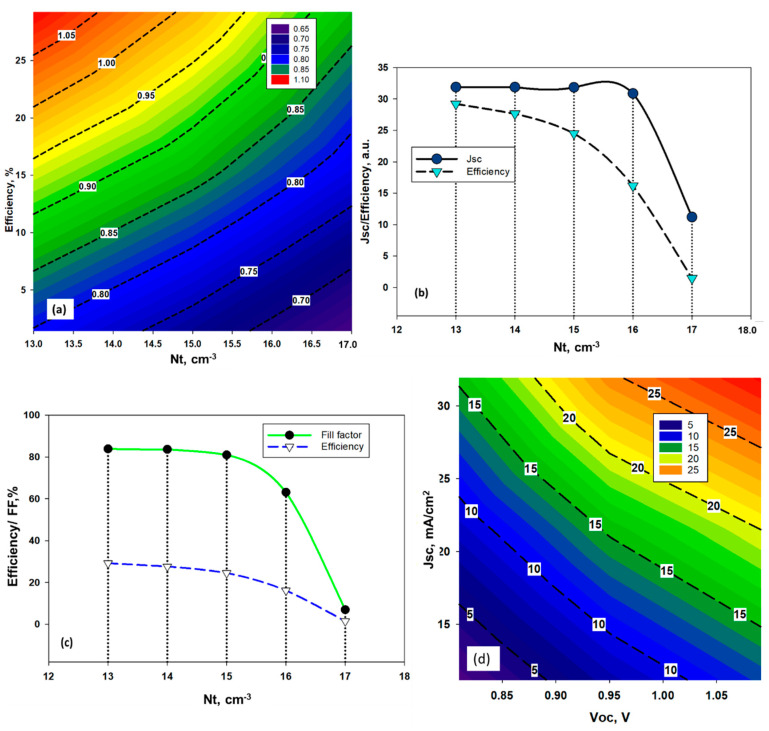
Effects of the variation of, (**a**) Efficiency and *Voc*, (**b**) *Jsc* and efficiency, and (**c**) *FF* and efficiency w.r.t. amphoteric defect density Nt, (**d**) relationship between *Voc* and *Jsc* w.r.t efficiency for different amphoteric defect states in CH_3_NH_3_SnBr_3_.

**Figure 4 nanomaterials-11-01218-f004:**
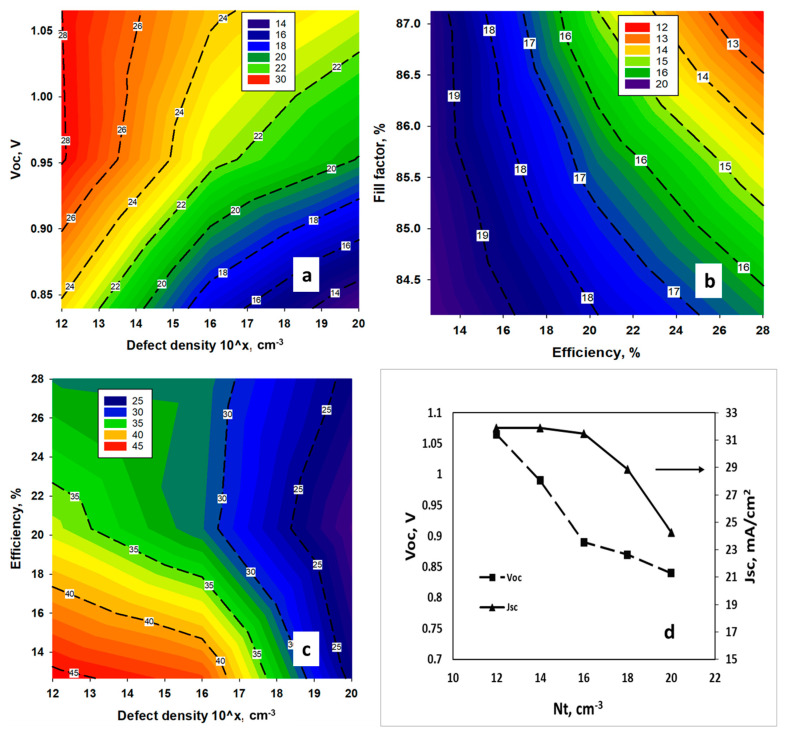
Effects of the variation of, (**a**) efficiency and *Voc*, (**b**) *Jsc* and efficiency and (**c**) *FF* and efficiency w.r.t. TiO2/Perovskite interface defect density Nt, (**d**) relationship between *Voc* and *Jsc* w.r.t various interface defect density in ETL/CH_3_NH_3_SnBr_3_ interface.

**Figure 5 nanomaterials-11-01218-f005:**
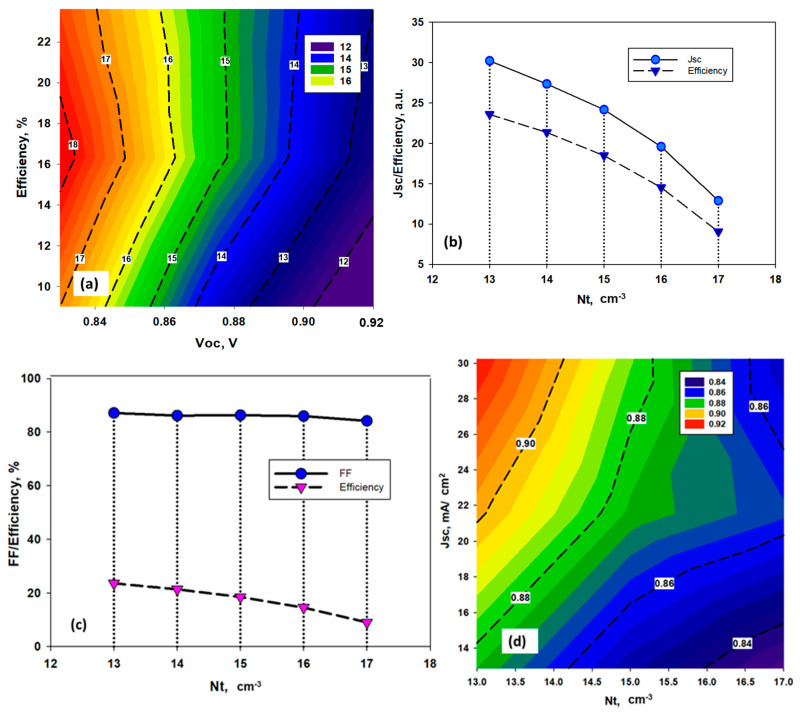
Effects of the variation of (**a**) Efficiency and *Voc*, (**b**) *Jsc* and efficiency and (**c**) *FF* and efficiency w.r.t. NiO/Perovskite interface defect density Nt (**d**) relationship between *Voc* and *Jsc* w.r.t various interface defect density in NiO/CH_3_NH_3_SnBr_3_ interface.

**Figure 6 nanomaterials-11-01218-f006:**
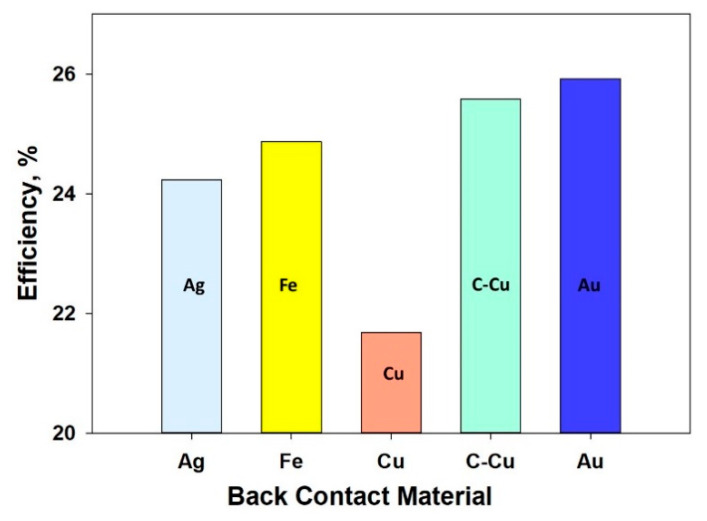
Solar cell efficiency w.r.t. different back contact metal.

**Figure 7 nanomaterials-11-01218-f007:**
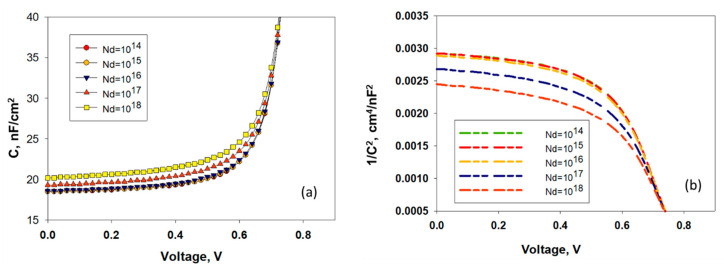
(**a**) C–V characteristics, and (**b**) Mott–Schottky plot attained using SCAPS as a function of shallow donor density (N_d_).

**Figure 8 nanomaterials-11-01218-f008:**
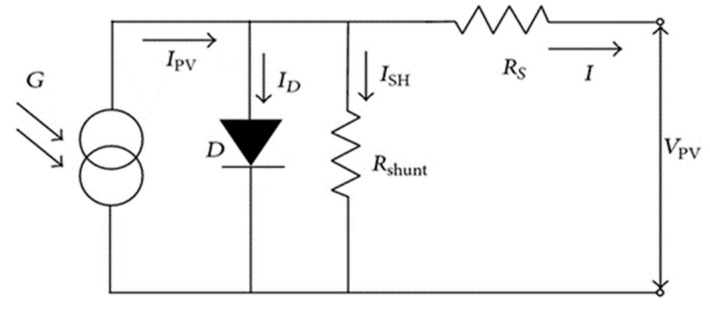
Equivalent circuit of a solar cell.

**Figure 9 nanomaterials-11-01218-f009:**
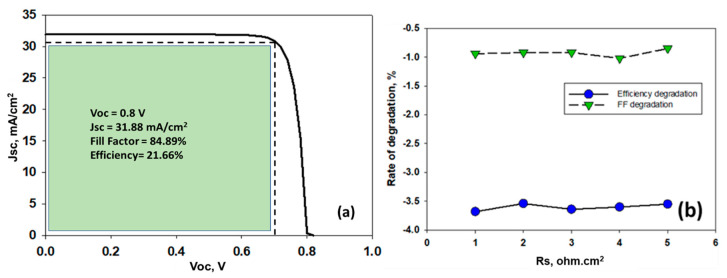
(**a**) Optimized CH_3_NH_3_SnBr_3_ based solar cell performance, (**b**) rate of change of *FF* and efficiency w.r.t Rs.

**Table 1 nanomaterials-11-01218-t001:** Simulation Parameter used in SCAPS-1D.

Parameter	FTO [35,36]	TiO_2_ [35,37]	CH_3_NH_3_SnBr_3_ [30,37]	NiO [34,37]
Thickness (µm)	0.4	0.05	0.5	0.15
*E_g_* (eV)	3.5	3.26	1.3	3.6
*χ* (eV)	4.0	4.2	4.17	1.8
*ε*	9.0	10.0	10.0	11.7
*N_C_* (cm^−3^)	2.2 × 10^18^	2.2 × 10^18^	2.2 × 10^18^	2.5 × 10^20^
*N_v_* (cm^−3^)	1.8 × 10^18^	1.8 × 10^18^	1.8 × 10^18^	2.5 ×10^20^
*μ_n_* (cm^2^/Vs)	20	20	1.6	2.8
*μ_p_* (cm^2^/Vs)	10	10	1.6	2.8
*N_D_* (cm^−3^)	1 × 10^19^	1 × 10^17^	1 × 10^13^	0
*N_A_* (cm^−3^)	0	0	1 × 10^13^	3 × 10^18^

**Table 2 nanomaterials-11-01218-t002:** Metal work function for different materials.

Back Contact Metal	Au	Ag	Fe	Cu	Cu Doped C
Metal work function (eV)	5.1	4.7	4.8	4.6	5.0

**Table 3 nanomaterials-11-01218-t003:** Effects of series resistance on photovoltaic parameters of CH_3_NH_3_SnBr_3_ based solar cell.

Resistance, Ohm.cm^2^	*V_oc_*, V	*J_sc_*, mA/cm^2^	*FF*, %	Efficiency, %
0	0.814	31.88	84.89	21.66
1	0.814	31.88	81.21	20.72
2	0.814	31.88	77.67	19.8
3	0.814	31.87	74.03	18.88
4	0.814	31.80	70.43	17.86
5	0.814	31.78	66.88	17.01

## Data Availability

The data will made available upon reasonable request to the corresponding author.

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
