# Peer review of "Defect Study and Modelling of SnX3-Based Perovskite Solar Cells with SCAPS-1D"

_nanomaterials, 2021, doi:10.3390/nano11051218_

Round 1
Reviewer 1 Report
Its a good paper, easy to read and scientifically sound.
Author Response
Thanks for reviewing the article.
Reviewer 2 Report
This paper presented the simulation of methyl ammonium tin bromide solar cells by SCAPS-1D. The results clearly exhibited the tolerance of defect density in the absorber layer and the interfaces are 1015 and 1014 cm-3, respectively. These simulated results are interesting to some researchers. I suggested that this paper can be accepted. However, the whole manuscript English should be revised carefully before publication. For example, there is incomplete sentence at line 278, etc.
Author Response
Dear Sir,
With due respect, I would like to express my gratitude to you for reviewing my article: Manuscript ID: nanomaterials-1154817. Based on your comments, I have modified/corrected/inserted all the necessary details and highlighted in the manuscript as well. I have put all the details as attached.

Reviewer 3 Report
The manuscript is presenting an interesting topic on the simulation of Sn-based perovskites. A number of parameters such as defect conc. and Nd and WF have been optimized. However,the few comments below are essential before we proceed with this publication:
-table I, the Eg of perovskite is 0.5 ev? but the band diagram shows it is 1.08 ev.
-it must be explained why rising the defect will reduce the Jsc and FF after concentration passed 10^14.
-citation is required to DOI: 10.1007/s11082-019-1802-3
-at the HTL defect is damaging Jsc. but not clear why. this could be well explained as it can be a guide for an experimentist.
-fig. 6 is not labeling the data.can't compare.
-why the V_bi is decreasing by increased voltage and C and Nt? I don't see this correlation with Fig. 7b thou.
when authors take my comments in and answer my Qs, i can reconsider my decision.
Author Response

(The authors gave the same response as above.)

Round 2
Reviewer 3 Report
my comments and questions were inserted properly and acceptable. i can accept it for publication now